# Characteristics of the Vertical and Horizontal Response Spectra of Earthquakes in the Jeju Island Region

**Jun-Kyoung Kim [1], Soung-Hoon Wee [2,*], Seong-Hwa Yoo [2] and Kwang-Hee Kim [3]**

[1] Department of Fire and Disaster Prevention, Semyung University, 65, Semyeong-ro, Jecheon-si 27136, Korea; kimjk3926@gmail.com
[2] Korea Institute of Geoscience and Mineral Resources (KIGAM), 124 Gwahang-ro, Yuseong-gu, Daejeon 34132, Korea; yoosh@kigam.re.kr
[3] Department of Geological Sciences, Pusan National University, 2 Busandaehak-ro 63 beon-gil, Kumjeong-gu, Busan 46241, Korea; kwanghee@pusan.ac.kr
* Correspondence: shwee@kigam.re.kr

**Abstract:** In this study, we evaluated the response spectra of 24 earthquake series, which includes 15 from the Kumamoto earthquake series and 9 from the Pohang earthquake series, and explored the effects of earthquake magnitude on the resonance frequencies of structures and buildings. Furthermore, the observations of this study were compared with the design response spectra, Regulatory Guide 1.60 (The United States Nuclear Regulatory Commission, 1973) for Korean nuclear power plants, and with the Korean Building Code (MOLIT, 2016, hereinafter referred to as KBC 2016) for general structures and buildings. The response spectra, after normalization with reference to the peak ground acceleration (PGA), were derived using a total of 423 horizontal and vertical accelerations. It was observed that the shapes of the horizontal and vertical response spectra were strongly dependent on the magnitude of the earthquake and the resonance frequency. Given the strong dependence of the response on the magnitude, it is suggested to consider magnitude > ML ~ 6.0 when establishing design response spectra. Compared to inland areas, a fairly higher amplitude of response at significantly lower frequency ranges could be attributed to the local geological environment of Jeju Island, which was formed by a surface volcano eruption and the distribution of unconsolidated Pleistocene marine sediments in the Jeju area. It is necessary to study the characteristic influence of layers with low shear wave velocity distributed in the Jeju region on seismic responses more rigorously while considering the frequency band and amplitudes at the surface of Jeju. The resonance frequencies of general low-rise and mid-rise buildings by the brief formula and those by design response spectra both suggested by KBC 2016 were overlapped, and these indicated that the seismic hazard could be much higher on general buildings in the Jeju region than in inland areas. Lastly, it is necessary to make the design standard criteria for Reg. Guide 1.60 and KBC 2016 more conservative in the lower frequency range of higher than 0.6 Hz and 2.0–6.0 Hz, respectively, which is significantly lower than those of the inland area, and to establish improved design response spectra with site-specific seismic design standards by referencing large amounts of qualitative data from the region around the Korean Peninsula.

**Keywords:** Gyeongju earthquake; horizontal response spectrum; KBC 2016; Regulatory Guide 1.60; resonance frequency; vertical response spectrum

## 1. Introduction

While recent studies have evaluated the characteristics of seismic response spectrum and seismic design mainly in the inland areas on the Korean Peninsula (Kim et al., 2016, Shin et al., 2016, Jee et al., 2018, Heo et al., 2018), none have studied seismic design in Jeju Island, the southernmost part of the Korean peninsula [1–4].

Areas surrounding Jeju island are seismically less active or fairly weak considering no large historical earthquakes (ML greater than 5.0) have been recorded in those regions.

However, seismic design is mandatory for major national infrastructures and utilities. Owing to the recent development of tourist facilities, a second airport is specifically being planned in Jeju island within a short period of time to cope with the rapid increase in the population. Furthermore, there is an increasing demand for large-scale utility facilities with large capacity such as nuclear power plants. Considering this, there has been an increasing need for studies evaluating the seismic design unique to this area. This study is the first to evaluate the seismic design in areas surrounding Jeju island. We investigated the response spectrum for the seismic design criterion according to the seismic characteristics of the Jeju area and compared the results with that of the seismic design of inland areas.

Jeju Island, located approximately 70 km (EW) and 30 km (NS) on the continental shelf south of the Korean peninsula, is a volcanic island with geochemical characteristics of oceanic island basalt (KIGAM, 2000), formed by the Pleistocene eruption [5]. Jeju is geologically much different from the inland areas. The island includes the Halla volcanic edifice in the central part, a lava plateau in the eastern part, and multiple cinder cones spread throughout the island (Tatsumi et al., 2005) [6]. A low-velocity layer (LVL) beneath the extrusive volcanic outcrops from the Quaternary age with depths of 100–200 m was identified using HVSR inversion by Kim and Hong (2012) [7]. Choi et al. (2007) also found that the LVL relates to the Pleistocene marine sediments of the Seoguipo and U formations, which in turn correspond to geological units specific to Jeju Island [8].

Seismic design characteristics are usually expressed as design response spectra, which include local/regional seismotectonic characteristics and seismic wave attenuations. In recent years, performance-based seismic design response spectra, which consider seismic performance based on the type and importance of the building structure, have been introduced and are an active area of research (Hahm, et al., 2012, Regulatory Guide 1.208, 2007) [9,10]. The KBC 2016 for general structures and buildings was revised in 2016, replacing KBC 2009. However, owing to the lack of studies on the seismotectonic characteristics of the Korean Peninsula, the suitability of the seismic design code is uncertain [9,10].

Researchers conducted studies on seismic response spectra and applied their work to seismic engineering. Furthermore, Housner (1959) developed seismic design spectra using eight horizontal ground motions that were recorded during four strong earthquakes [11]. Many others have researched the response spectra. Bozorgnia and Campbell (2004) and Elgamal and He (2004) stated that the vertical and horizontal response spectra are significantly influenced by factors such as resonance frequency, epicentral distance, soil type, earthquake magnitude, and fault movement [12,13].

Kim and Oh (2016), Shin et al. (2016), Kim et al. (2016), Heo et al. (2018), Kim et al. (2018), and Kim et al. (2019) studied the response spectra for inland seismic stations based on the Fukuoka, Gyeongju, and Pohang earthquake series [1,2,4,14–16].

Therefore, the verification of seismic design standards for the Jeju area is of particular interest owing to the few recent occurrences of macro earthquakes larger than ML~4.0. The Kumamoto and Pohang earthquake series have provided many ground motions larger than $M_L$~6.0 to seismic stations in the Jeju area.

The results of this study were compared to the standard response spectra for nuclear power plants and related facilities specified in the Regulatory Guide 1.60 (United States Nuclear Regulatory Commission, 1973) and the KBC 2016 (MOLIT, 2016) guidelines that address the design response spectra for general structures and buildings [17,18]. Furthermore, we used the KBC 2016 (MOLIT, 2016) to assess the suitability of the two principal seismic design standards used in Korea to the Jeju region [18].

## 2. Ground Motions

In 2016, 15 Kumamoto earthquake series observed in the most seismic stations in Jeju caused over 48 casualties. Foreshocks, the main shock, and aftershocks occurred continuously from 16th–19th April in the Kumamoto area, in the central part of Kyushu Island, southwest Japan.

Table 1 shows the distances from the earthquake epicenter to the 11 seismic stations in the range of 367–429 km in the Jeju area, which is similar to the distances from the epicenter to all seismic stations in this study. Focal depths of 10–20 km (typical mid-crustal depths) were observed for the Kumamoto earthquake series. In this case, 15 macro earthquakes series with magnitudes ranging from 4.8 to 7.3 in ML in a narrow zone of epicenters were felt over a very wide area and affected buildings and structures both in the Jeju area and some inland cities within a short period of time. Owing to the uniformity of the propagation paths of each earthquake group to seismic stations in Jeju, it is possible to reduce heterogeneous effects caused by the propagation path.

**Table 1.** Dates and seismic stations of earthquake occurrences.

| No. | Event Date | Lat. | Log. | $M_L$ | Stations | Remarks |
|---|---|---|---|---|---|---|
| 1 | 2016-04-14 21:26:00 | 32.70 | 130.80 | 6.5 | GOS1,HALB,JJB,JJU1,MRD,SGP1,SSP | Kumamoto Event |
| 2 | 2016-04-14 22:07:00 | 32.80 | 130.80 | 5.7 | GOS1,HALB,JJB,JJU1,MRD,SGP1,SSP | Kumamoto Event |
| 3 | 2016-04-14 22:38:00 | 32.70 | 130.70 | 5.0 | GOS1,HALB,JJB,JJU1,MRD,SGP1,SSP | Kumamoto Event |
| 4 | 2016-04-15 00:03:00 | 32.70 | 130.80 | 6.4 | GOS1,HALB,JJB,JJU1,MRD,SGP1,SSP | Kumamoto Event |
| 5 | 2016-04-15 01:53:00 | 32.70 | 130.80 | 4.8 | GOS1,HALB,JJB,JJU1,MRD,SGP1,SSP | Kumamoto Event |
| 6 | 2016-04-16 01:25:00 | 32.80 | 130.80 | 7.3 | GOS1,HALB,JJB,JJU1,MRD,SGP1,SSP | Kumamoto Event |
| 7 | 2016-04-16 01:45:00 | 32.90 | 130.90 | 6.0 | GOS1,HALB,JJB,JJU1,MRD,SGP1,SSP | Kumamoto Event |
| 8 | 2016-04-16 03:03:00 | 33.00 | 131.10 | 5.8 | GOS1,HALB,JJB,JJU1,MRD,SGP1,SSP | Kumamoto Event |
| 9 | 2016-04-16 03:55:00 | 33.00 | 131.20 | 5.8 | GOS1,HALB,JJB,JJU1,MRD,SGP1,SSP | Kumamoto Event |
| 10 | 2016-04-16 07:23:00 | 32.80 | 130.80 | 4.8 | GOS1,HALB,JJB,JJU1,MRD,SGP1,SSP | Kumamoto Event |
| 11 | 2016-04-16 09:48:00 | 32.90 | 130.80 | 5.4 | GOS1,HALB,JJB,JJU1,MRD,SGP1,SSP | Kumamoto Event |
| 12 | 2016-04-16 16:01:00 | 32.80 | 130.80 | 5.3 | GOS1,HALB,JJB,JJU1,MRD,SGP1,SSP | Kumamoto Event |
| 13 | 2016-04-17 00:14:00 | 33.00 | 131.10 | 4.9 | GOS1,HALB,JJB,JJU1,MRD,SGP1,SSP | Kumamoto Event |
| 14 | 2016-04-19 17:52:00 | 32.60 | 130.70 | 5.5 | GOS1,HALB,JJB,JJU1,MRD,SGP1,SSP | Kumamoto Event |
| 15 | 2016-04-19 20:47:00 | 32.60 | 130.70 | 5.0 | GOS1,HALB,JJB,JJU1,MRD,SGP1,SSP | Kumamoto Event |
| 16 | 2017-11-15 14:29:31 | 36.11 | 129.37 | 5.4 | CJD,JJU2,SGP2,UDO | Pohang Event |
| 17 | 2017-11-15 14:32:59 | 36.10 | 129.36 | 3.6 | CJD,JJU2,SGP2,UDO | Pohang Event |
| 18 | 2017-11-15 15:09:49 | 36.09 | 129.34 | 3.5 | CJD,JJU2,SGP2,UDO | Pohang Event |
| 19 | 2017-11-15 16:49:30 | 36.12 | 129.36 | 4.3 | CJD,JJU2,SGP2,UDO | Pohang Event |
| 20 | 2017-11-19 23:45:47 | 36.12 | 129.36 | 3.5 | CJD,JJU2,SGP2,UDO | Pohang Event |
| 21 | 2017-11-20 06:05:15 | 36.14 | 129.36 | 3.6 | CJD,JJU2,SGP2,UDO | Pohang Event |
| 22 | 2017-12-25 16:19:22 | 36.11 | 129.36 | 3.5 | CJD,JJU2,SGP2,UDO | Pohang Event |
| 23 | 2018-02-11 05:03:03 | 36.08 | 129.33 | 4.6 | CJD,JJU2,SGP2,UDO | Pohang Event |
| 24 | 2019-02-10 12:53:38 | 36.16 | 129.90 | 4.1 | CJD,JJU2,SGP2,UDO | Pohang Event |

In particular, JJU1 and SGP1 were transferred to JJU2 and SGP2, respectively, within approximately 200 m owing to technical problems such as noise level after the occurrence of the Kumamoto earthquake. In addition, the SSP was closed after the occurrence of the earthquake, and a new UDO was installed on a nearby island. Observations at UDO are being continued to this day.

Therefore, considering the Kumamoto earthquake series was not recorded at JJU2 and SGP2, and at the newly installed UDO, the seismic response spectra were investigated using the ground motions of the 2017 Pohang earthquake series (Figure 1 and Table 1) with ML between 3.5–5.4, which occurred after the Kumamoto earthquake series. We compared the seismic response spectra of JJU2 close to JJU1 and SGP2 station close to SGP1. In this study, we evaluated 423 horizontal and vertical ground motions from 15 Kumamoto earthquake series observed at 7 stations and 9 Pohang earthquake series observed at 4 stations (Tables 1 and 2).

Table 1 shows the date and time of the occurrence of the earthquake's epicenter, the geographical location of the 15 Kumamoto and 9 Pohang earthquakes, and the station names at which ground motions were recorded. The 423 ground motions (horizontal and vertical components) were processed by 5% cosine tapering and linear trend correction. The damping ratios of general buildings and structures range from 3 to 7%. Similar to previous studies, the response spectrum analysis was chosen as 5%. The data comprised of acceleration records that was used to review the response spectra. Figure 1 shows the geographical locations of the 15 Kumamoto (small red circle) and 9 Pohang (large red circle) earthquake series and 11 seismic stations (green triangles). Figures 2 and 3 shows the time history waveforms of the mainshocks and aftershock for the Kumamoto at JJB station and Pohang earthquake series at JJB2 station, respectively.

The duration of ground motions was considered to be 300 s, and the sampling interval for the acceleration of ground motions was 0.01 s. Although some aspects of the S-wave, such as the maximum amplitude and duration, are influenced by the magnitude and distance from the epicenter (Table 2), all ground motions had the same duration.

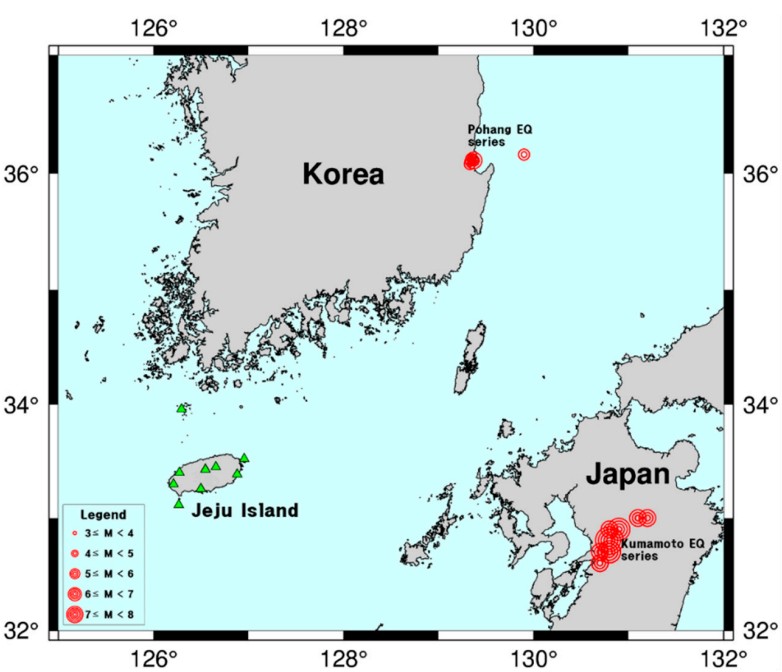

**Figure 1.** Locations of the Kumamoto earthquake series (red circles), Pohang earthquake series (red circles), and the 11 seismic stations (green triangles).

**Table 2.** Seismic stations and their distances from the epicenter of the main event.

| No. | Station | Lat. | Long. | Sensor | Distances |
|-----|---------|------|-------|--------|-----------|
| 1 | GOS1 | 33.3003 | 126.2100 | ES-T | 431.65 |
| 2 | HALB | 33.4019 | 126.2729 | ES-DH | 427.20 |
| 3 | JJU1 | 33.4306 | 126.5463 | ES-T | 402.53 |
| 4 | JJU2 | 33.4294 | 126.5463 | ES-T | 394.14 |
| 5 | SGP1 | 33.2587 | 126.4994 | ES-T | 404.40 |
| 6 | SGP2 | 33.2587 | 126.4983 | ES-T | 411.63 |
| 7 | SSP | 33.3873 | 126.8801 | ES-T | 371.16 |
| 8 | CJD | 33.9594 | 126.2934 | ES-T | 368.24 |
| 9 | UDO | 33.5228 | 126.954 | ES-T | 362.50 |
| 10 | MRD | 33.1166 | 126.2659 | ES-T | 424.94 |
| 11 | JJB | 33.4515 | 126.6559 | ES-DH | 392.85 |

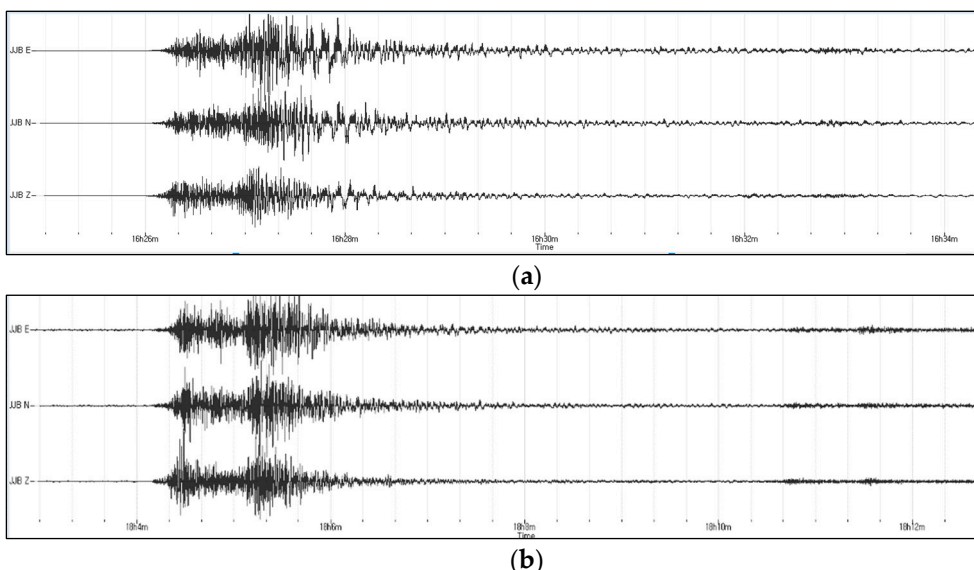

**Figure 2.** Three-component time history waveforms (EW, NS, UD) of the main earthquake (M, 7.3) and aftershock (M, 5.7) among the Kumamoto earthquake series at JJB station are shown in (**a**,**b**), respectively.

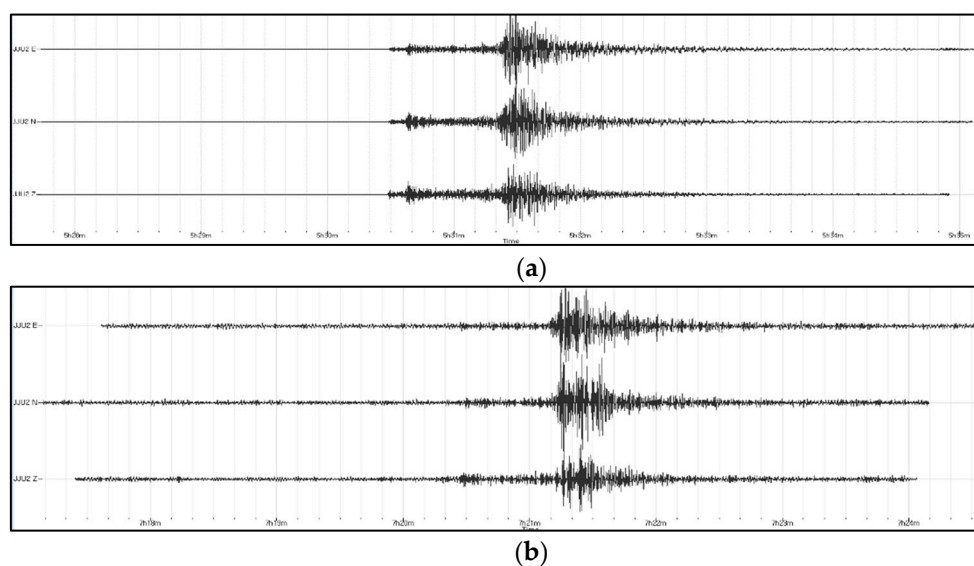

**Figure 3.** Three-component time history waveforms (EW, NS, UD) of the main earthquake (M, 5.4) and aftershock (M, 3.5) among the Pohang earthquake series at JJU2 station are shown in (**a**,**b**), respectively.

## 3. Frequency Bands and Normalisation of the Response Spectra

The acceleration response spectra in this study can be given as follows:

$$S_a(\omega_n, \zeta) = \max_t \left| \ddot{u}^t(t) \right| = \max_t \left| \ddot{u}^t(t) + \ddot{u}_g(t) \right| = \max_t \left| -2\zeta\omega_n\dot{u}(t) - \omega_n^2 u(t) \right| \quad (1)$$

where $S_a(\omega_n, \zeta)$ is the acceleration response spectrum, $\ddot{u}^t(t)$ is the relative acceleration, $\dot{u}(t)$ is the relative velocity, $\zeta$ is the damping value, and $\omega_n$ is the angular velocity.

We calculated the responses for frequencies ranging from 0.1–50 Hz, at intervals of 0.1. It was observed that the ground motions of the high-frequency bands (>30 Hz) attenuated rapidly as it traveled through the surface and subsurface, and exceeded the resonance

periods of most building structures, but not the attached components, thereby posing a minimal threat.

Although the Reg. Guide 1.60 (which sets the standards for nuclear power plant construction) states that the range of the responses is only up to 33 Hz, we calculated and compared the responses up to 50 Hz while considering a safety margin. While the resonance period was unique for each structure, most buildings exhibited resonance frequencies with wide ranges, based on the height and structure of the buildings. Therefore, it is essential to consider the characteristics of building structures in this band.

Considering the maximum accelerations of ground motions depend mainly on the epicentral distance, the magnitude of the earthquake, and the site profile, the response spectra of each motion should be normalized to the peak ground acceleration (PGA) or another norm. Although three normalization methods have been developed (involving PGA, effective peak acceleration, and spectral intensity), PGA is the most commonly used. This study used the PGA normalization method generally adopted by NUREG-0800(1975), Regulatory Guide 1.165 (USNRC, 1997), and 1.208 (USNRC, 2007) [10,19,20]. Therefore, all response spectra mentioned in the following sections indicate the normalized response spectra using PGA.

## 4. Characteristics of the Horizontal Response Spectra in Jeju Island

Figure 4 shows the horizontal response spectra of 7 stations (Table 2), including GOS, HALB, JJB, JJU1, MRD, and SGP1, for the 15 Kumamoto earthquake sequence (ML: 4.8–7.3). It can be seen that as the frequency increases in the low-frequency band, the response gradually increases. The response reaches a peak value in the mid-frequency band and finally decreases in the high-frequency band, as suggested by various seismic design standards (Reg Guide 1.60, 1.165, and 1.208 (US NRC 1973, 1997, 2007) and ASME (Gupta and Hall, 2017) [10,17,20,21]. Particularly, the responses of GOS, HALB, MRD, and SSP exhibited a characteristic section where the slope was remarkably gentle in the frequency range of 0.3–0.5 Hz.

**(a)**
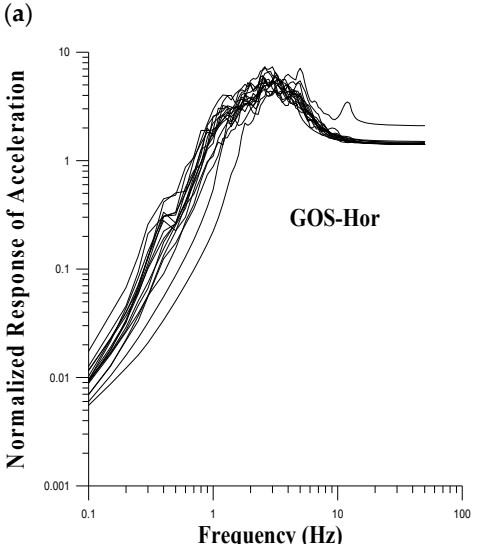

**(b)**
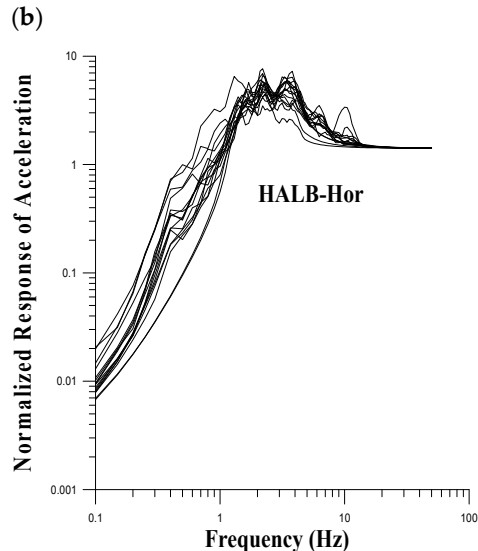

**Figure 4.** *Cont.*

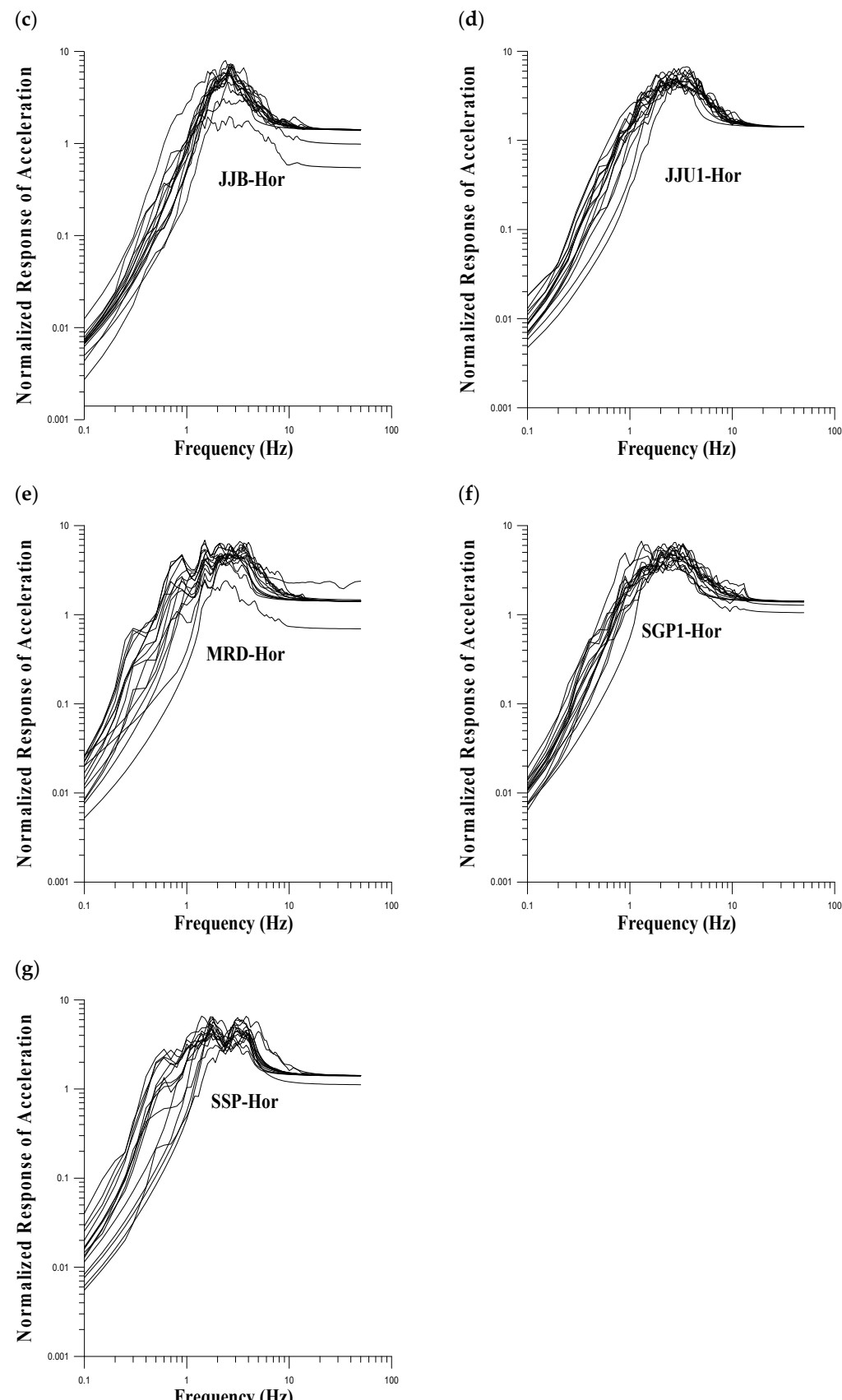

**Figure 4.** Normalized horizontal response spectra at (**a**) GOS, (**b**) HALB, (**c**) JJB, (**d**) JJU1, (**e**) MRD, (**f**) SGP1, and (**g**) SSP using Kumamoto earthquakes series.

It is necessary to consider ground motion data as much as possible to determine a response spectrum that was more representative of the Jeju region. Therefore, we included four additional stations, that is, CJD (northwest area), JJU2 (central), and SGP2 (northeast), and UDO (northeast island) (Table 2). JJU2 and SGP2 replaced JJU1 and SGP1 and the newly installed UDO after the Kumamoto earthquake. In particular, although CJD was installed before the Kumamoto earthquake, no data are available on that particular event.

Therefore, to investigate the response spectrum of the CJD, JJU2, SGP2, and UDO stations, we considered 9 Pohang earthquake sequences (ML: 3.5–5.4), which occurred in 2017 on the inland Korean Peninsula.

Figure 5 shows individual horizontal response spectra of CJD, JJU2, SGP2, and UDO. CJD exhibited a very large variability in response; the amplitude of response spectra decreased with a decrease in the earthquake magnitude. Similarly, JJU2 was characterized by very high variability. SGP2 also showed a large response at ML 5.4, whereas some showed responses that were abnormally very far from most responses in the high frequency bands. Although UDO showed fairly large variability in the high frequency band, it was comparatively lower than that of the other stations in the entire frequency band. Therefore, all three stations except UDO were excluded owing to their relatively large variability compared to the previous seven stations. As a result, the responses of the remaining 8 stations were investigated to represent the Jeju area.

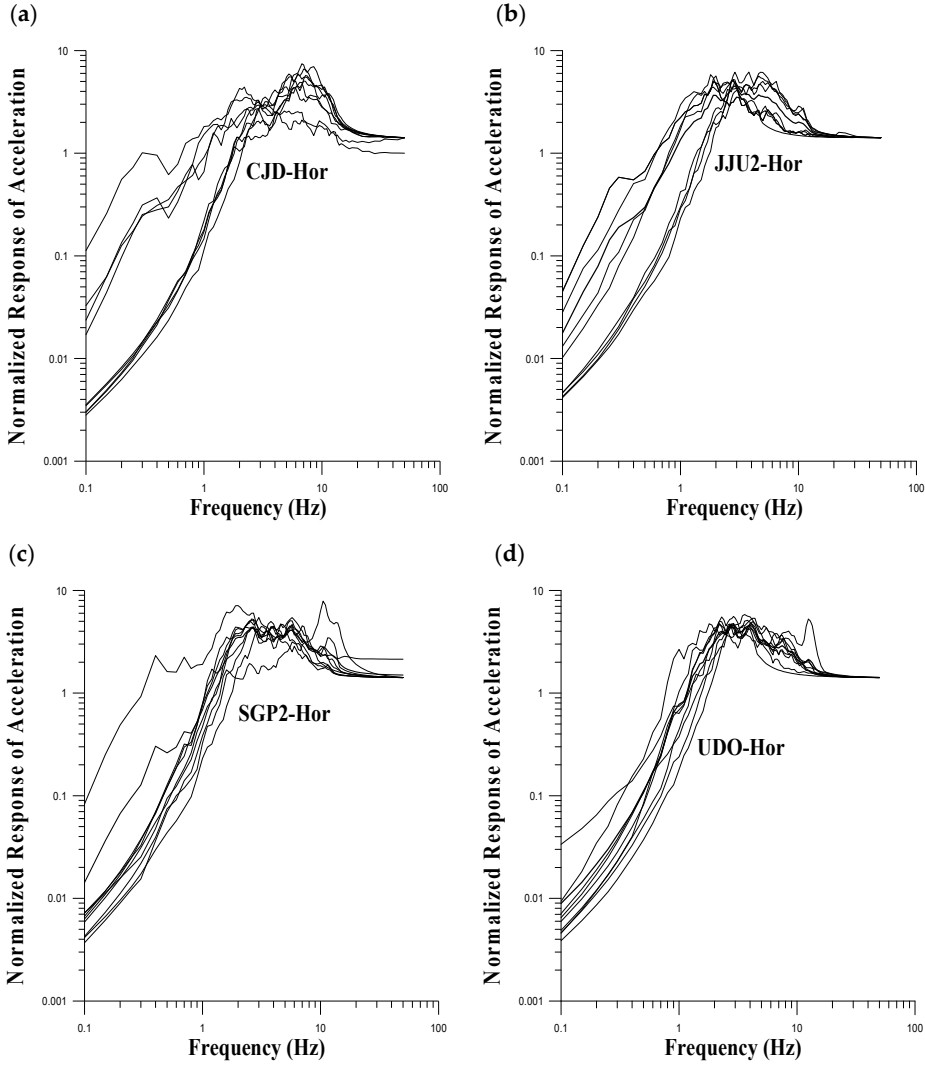

**Figure 5.** Normalized horizontal response spectra at (**a**) CJD, (**b**) JJU2, (**c**) SGP2, and (**d**) UDO using Pohang earthquakes series.

The Pohang earthquake series (Table 1) exhibited an ML of 5.4, with three earthquakes in the ML 4 level (4.6, 4.3, and 4.1) and five in the ML 3 level. In contrast, seismic ground motions from the Kumamoto earthquake (table x) can be classified into three groups: Four with ML greater than 6.0 (7.3, 6.4, 6.3, and 6.0), six with ML greater than 5.3 (5.8, 5.8, 5.7, 5.5, 5.4, 5.3), and five with ML greater than 4.8 (5.0, 5.0, 4.9, 4.8, 4.8). It can be seen that a very large difference exists in the total numbers of events (15 in Kumamoto and. 9 in the Pohang series) and the number of events with ML greater than 5.0 (12 in Kumamoto and 1 in the Pohang series). On comparing the seismic energy release, the largest earthquake (ML 7.3) in the Kumamoto series was found to have approximately 708 ($10^{(1.5 \times 1.9)}$) times the seismic energy (ML 5.4) of the Pohang sequence, indicating a large asymmetry of the distributions of the earthquake scale.

Since the above four stations are permanent and included in national networks, devices such as sensors and recorders and the site effects of the stations need to qualify a certain level of national quality assurance. Furthermore, owing to the intra-plate environments of the paths from the Kumamoto and Pohang earthquake epicenters to the stations in Jeju, the elastic and inelastic properties of regional crustal propagations are also considered to be the same (Silva et al., 1999) [22].

Assuming there are no significant differences in the site effect and the propagation path between Pohang and Kumamoto series, we can say that the difference in magnitude of the largest event, number of events with ML greater than 6.0, and their distribution related to seismic sources among three factors caused the largest influences on the seismic energies, which were sequentially reflected in the response spectrum.

Therefore, this suggests the possibility of improving the reliability of the response spectra by reducing the variability representing an arbitrary region by securing a sufficient seismic database, including earthquakes with a fairly large magnitude (ML 6.0) and an increased number of larger events.

## 5. Results and Discussion

### 5.1. Comparison of the Response Spectra between Two Pairs of Stations: JJU1 and JJU2, and SGP1 and SGP2

To further investigate the causes of the large variability in the response spectra of the four stations, we compared the responses of two stations located close to each other under similar site effect conditions, which helps simplify and focus on the other two factors, considering the influence of the site effect among the three factors affecting the response spectra becomes negligible.

On 16th December 2016, the Korea Meteorological Administration (KMA) closed JJU1 and moved to JJU2, which was at a distance of approximately 200 m from JJU1. Similarly, SGP1 was closed and moved to SGP2, which was approximately 200 m from SGP1. We compared the responses of JJU1 and SGP1 for the Kumamoto series and those of JJU2 and SGP2 for the Pohang series.

Figure 6a,b show the horizontal responses of JJU1 and JJU2, and SGP1 and SGP2, before and after relocation. Compared to the responses of JJU1 and SGP1, the results of JJU2 and SGP2 showed a significantly higher variability for all frequency bands. Assuming the movement was not only within the ~200 m distance, and that the conditions of the new site generally improve through ground surveys, the difference of the site effect on the responses before and after relocation can be neglected.

In addition, the wave propagation paths from the Kumamoto and Pohang earthquakes to the Jeju region exhibit the same intra-plate tectonic characteristics (Silva et al., 1999) [22]. Since similar tectonic environments show similar elastic and inelastic properties of wave propagation, the difference in the ground motions reflected in the response spectra of the two paths is almost negligible. Considering this, the increased variability between the before and after movement suggests that seismic sources are most significantly involved. The importance of earthquake magnitude, which is one of the components of the seismic sources, is confirmed again in addition to the previous section.

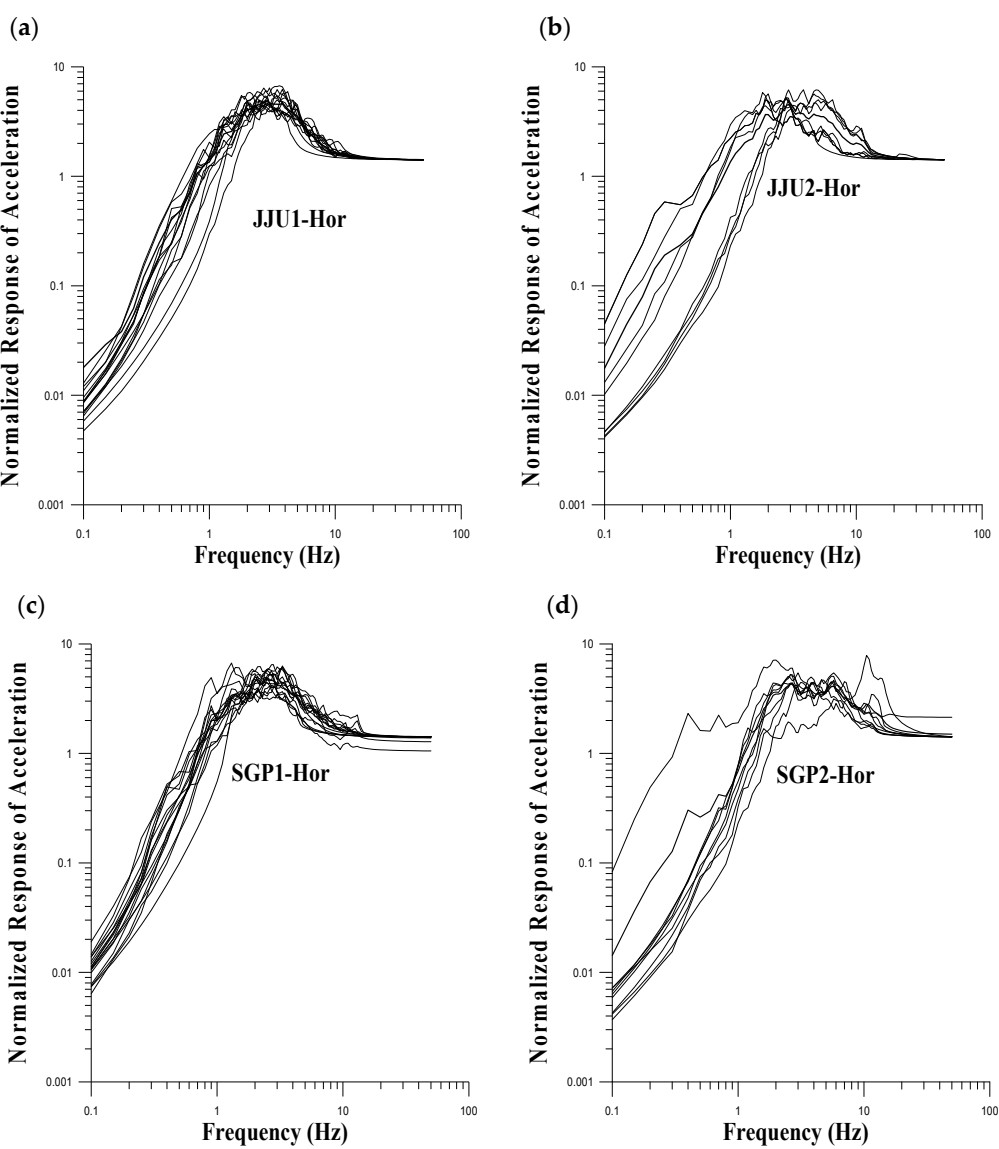

**Figure 6.** Normalized horizontal response spectra at (**a**) JJU1, (**b**) JJU2, (**c**) SGP1, and (**d**) SGP2 using Kumamoto and Pohang earthquakes series.

*5.2. Influence of Earthquake Magnitude on the Horizontal and Vertical Response Spectra*

We investigated the dependency of earthquake magnitude, one of the most important components of seismic sources, by grouping earthquake magnitudes according to their magnitude levels: ML 6.0–7.3 (4 earthquakes), ML 5.3–5.8 (6 earthquakes), and between ML 4.6–5.0 (5 earthquakes). Accordingly, we investigated the influence of earthquake magnitude on the horizontal response spectra.

Furthermore, we extended the investigation of dependence on the magnitude to all eight stations and events. Similar to the previous case, all events were classified into three levels based on the earthquake magnitude. Based on the magnitude distribution of the 15 Kumamoto earthquakes, the earthquakes in this study were divided into three groups according to the earthquake magnitude: ML 6.0–7.3 (4 earthquakes), ML 5.3–5.8 (6 earthquakes), and ML 4.6–5.0 (5 earthquakes).

Figure 7a shows three horizontal responses for the three magnitude levels. The responses increased as the magnitude of the earthquake increased in the frequency band below ~1 Hz. Further, no significant difference was observed among the responses of the

three groups in the frequency band of 1–9 Hz. Lastly, all three responses decreased rapidly in the frequency band above ~9 Hz and converged to similar values.

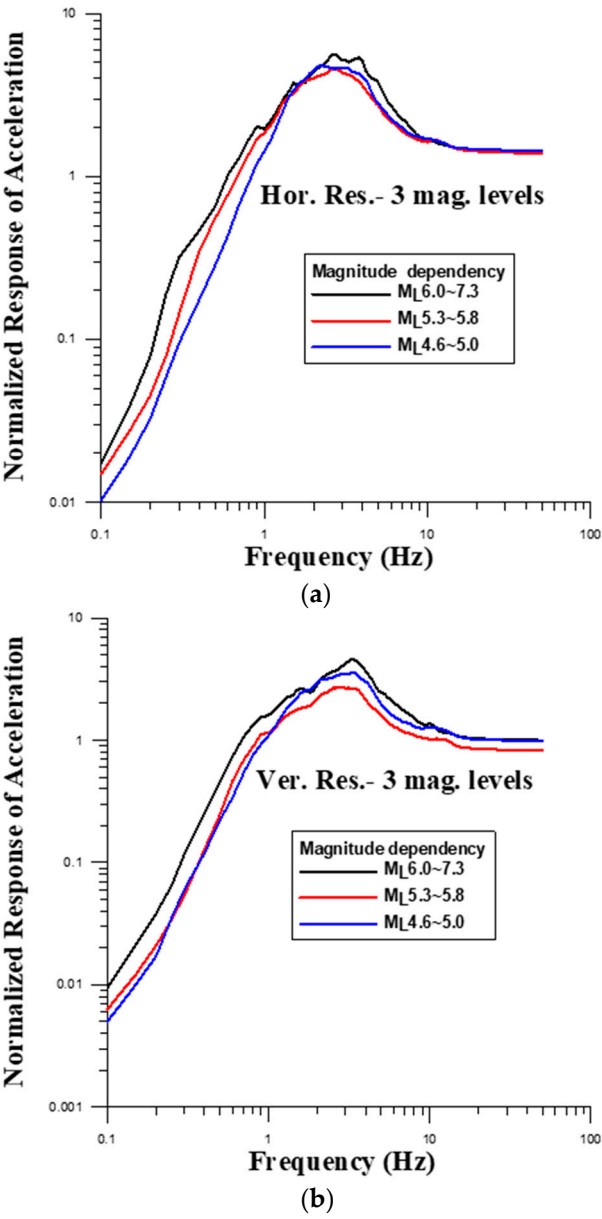

**Figure 7.** Normalized (**a**) horizontal and (**b**) vertical response spectra of three groups of earthquake magnitude (15 events): $M_L$ 6.0~7.3 (black solid), 5.3~5.8 (red solid), and 4.6~5.0 (blue solid), respectively.

The amplitudes and locations of the maximum responses for the first, second, and third groups were ~4.9 (3.3 Hz), 4.6 (~2.6 Hz), and ~4.6 (~2.2 Hz), respectively. The maximum responses and corresponding frequencies were in the range of 4.6–4.9 and 2.2–3.3 Hz, respectively, indicating no significant differences in the amplitudes and frequency bands of the groups. Therefore, the dependence of the response spectrum on the earthquake magnitude is limited to the low-frequency band below ~ 1 Hz.

According to the extended and point seismic source models (Brune, 1970, Madariaga, 1976, Beresnev and Atkinson, 1998, Boore, 2003, Motazedian and Atkinson, 2015), earthquakes of ML between 6.0 and 7.3 exhibit a seismic source spectrum much higher in amplitude and broader in frequency than those of earthquakes of smaller ML [23–27]. In addition, as the epicenter distance increased, the seismic energy in the higher and lower

frequencies decreased rapidly and gradually, respectively, and then those of lower frequencies propagated farther. Therefore, as the epicentral distance increases, larger earthquakes exhibit higher responses in lower frequency bands, as shown in Figure 7a,b), and as suggested by previous studies (Yang and Lee, 2007, Campbell and Bozorgnia, 2014, Kim et al., 2018, Kim et al., 2019) [15,16,28,29].

Owing to the similarities in the effects of the site and wave propagation path (Silva et al., 1999) on the shapes of the response spectra, the seismic source, that is, the distribution of larger magnitude, is essential in determining the characteristics of the response spectrum, especially in the low-frequency region, as suggested in previous studies (Yang and Lee, 2007, Campbell and Bozorgnia, 2014, Kim et al., 2018, Kim et al., 2019), including recent studies (Bindi, et al., 2017, Douglas; and Philippe, 2011) [15,16,28–31]. When establishing the response spectra representing an arbitrary region while compensating for the conservativeness in the low-frequency band, it is necessary to consider a number of fairly large magnitudes of earthquakes (e.g., ML 6.0 or larger).

The historical earthquake records show over 10 earthquakes of magnitude 6.0 or greater in the Korean Peninsula. However, no earthquakes of over 6.0 magnitude have occurred in the Korean Peninsula in recent times. Thus, to obtain seismic ground motions with a potential magnitude of over 6.0, which can potentially impact the Korean Peninsula, it is necessary to collect seismic data of magnitude 6.0 or larger from the southeastern region of Japan, considering it is geographically closer to the Korean Peninsula and earthquakes are a frequent occurrence.

For the vertical component, we investigated the influence of magnitude on the average response spectra of all eight stations using the entire event. Figure 7b shows the shape characteristics of the vertical response spectra depending on the magnitude, which was similar to the horizontal response spectra in the low-frequency band, although the vertical response of the 2nd group in the low-frequency band was much closer to that of the group with earthquakes having the smallest magnitudes (ML 4.6–5.0). However, the response of the 1st and 3rd groups showed a distinct difference, indicating that the vertical response spectra were dependent on the magnitude.

The dependence of the response spectra on epicentral distance suggested by many studies (Yang and Lee, 2007, Kim et al., 2018, Kim et al., 2019) was not investigated in this study because the distances to all the stations of Jeju from epicentres of two main earthquake sequences (Table 1) were in the range of ~360–460 km (at most ~100 km difference); this distance among stations was too small to investigate the dependence on epicentral distances [15,16,28].

*5.3. Horizontal and Vertical Response Spectra of the Jeju Region*

Individual, mean, and mean plus 1σ response spectra of 8 stations and all the events for horizontal and vertical components in Jeju are shown in Figure 8a,b. Since Jeju island is approximately 74 km long and 32 km wide in roughly east-west and north-south directions, respectively (Figure 1), 8 stations can be considered to be fairly well representative of characteristics of response spectra for the Jeju island (9.25 km long and 4.0 km wide per each station).

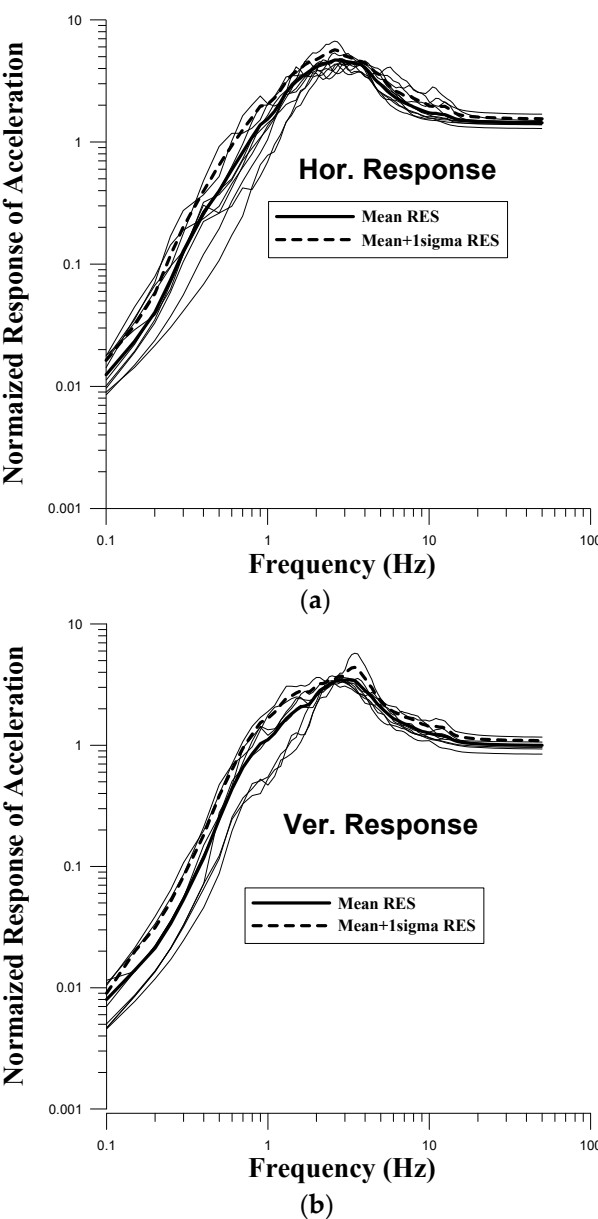

**Figure 8.** Mean and mean +1 σ, (**a**) horizontal and (**b**) vertical response spectra of all the events and 8 stations.

The mean plus 1σ horizontal response spectra (Figure 8a) increases gradually in the low frequency band, reached a maximum of ~5.7 at 2.6 Hz, and gradually decreases and converges to ~1.6 at high frequencies. Many regulatory design criteria, including Reg. Guide 1.60, 1.165, 1.208 (US NRC, 1973, 1997, 2007), and ASME (Gupta and Hall, 2017) adopt the mean plus 1σ values to ensure the conservatism of the seismic design response spectrum [10,17,20,21].

In recent years, many studies have investigated the response spectra of inland Korea, except Jeju Island, using ground motions Fukuoka events (Kim and Oh, 2016), Pohang events series (Kim et al., 2018), and the Gyoungju events (Kim et al., 2019) [14–16]. These previous studies (conducted 2014 onward) state that the horizontal response spectra commonly exhibit peak values ranging from 4.5 to 5.0 at frequencies of 9 Hz or higher. In contrast, the maximum horizontal response in the Jeju area is ~5.6, which is a fairly higher (12–24 %) than that of the inland area, whereas the frequency band with very high responses ranges between 2.0 and 6.0 Hz, which is significantly lower than that of the inland areas.

The higher responses including maximum amplitude formed in the lower frequency band could be attributed to the local geological environment of Jeju island, which was formed by the eruption of a surface volcano with lower shear velocity in contrast to the inland granite-based bedrock having higher shear wave velocity (Kim and Hong, 2012) [7]. This could also be attributed to the existence of unconsolidated Pleistocene marine sediments distributed widely in the Jeju area, which were identified by penetrating well data (Oh et al., 2000) having a remarkably low shear wave velocity at ~200 m under the surface [32]. The layer was also identified by the inversion of the HVSR and dispersion curves (Kim and Hong, 2012) and the deep resistivity sounding having relatively higher conductivity (<100 $\Omega$ m) compared to that of the over- and underlying layers (Choi et al., 2007) [7,8].

Owing to the larger contrast of elastic impedance (shear velocity multiplied by density) among adjacent layers, the higher responses, including the maximum amplitude, can be developed in a frequency band much lower than that of the inland areas. Considering the general relationship between the resonant frequency and the thickness and shear wave velocity of the layers, the resonant frequency of the site decreases as the velocity of the layers with an assumed thickness decreases (Yoon et al., 2006, Sun et al., 2007) [33,34].

Furthermore, the amplitude of the maximum response in Jeju being 12–24% higher than that of land can be attributed to the greater elastic impedance contrast between the strata owing to the existence of low-velocity layers distributed widely in Jeju. In the near future, it would be necessary to thoroughly investigate the influence of low shear wave velocity layers, surface volcano eruption, and marine sediments in depth on the seismic response spectra with frequency bands and amplitudes.

Considering a simple formula, let To = 0.1 N (equation 0306.5.6) be the resonance frequency of the buildings suggested by KBC 2016, where N indicates the number of floors. According to the formula, the natural resonance frequencies of low-rise and mid-rise buildings are usually ~5 Hz (2-story), 3.3 Hz (3-story), 2.5 Hz (4-story), and 2 Hz (5-story), when the formula (0306.5.6) or formula (0306.5.5) of KBC2016 code is applied. Most frequency ranges with very high responses (~ 2.0–6.0 Hz), including the maximum in this study, overlap with the natural resonance frequencies of general low-rise and mid-rise buildings. In contrast, the frequency bands exhibiting the peak values of the inland response spectrum are higher than 9 Hz, which is far from that of general buildings.

Therefore, considering the relatively large (12–24%) amplitude of the peak response and the resonant frequency band overlapping frequencies of buildings and structures in Jeju, the seismic hazard could be higher on general buildings than on inland structures. A more detailed comparison with the KBC 2016 design standard will be discussed in a later section.

*5.4. Ratio of the Vertical Response Spectra to the Horizontal Response Spectra (V/H)*

The horizontal and vertical standards are commonly presented simultaneously, considering the vertical response standard is essential in seismic design. Many seismic design criteria (Table 3) include the vertical to horizontal (V/H) response spectra ratio over short and long periods.

Figure 9 shows the average and individual V/H values of the 8 stations. The V/H was higher than 2/3 for most frequency bands except 0.3–0.4 Hz and ~2 Hz, which were consistent with the V/H scales suggested by various seismic design standards (Table 3) and a previous study (Bozorgnia and Campbell, 2004) [12]. Although the frequency range of high responses including the maximum has a higher amplitude than that of inland areas, it shows similar characteristics to inland H/V, indicating that the vertical component response value increases simultaneously as the horizontal component response value increases.

**Table 3.** The vertical-to-horizontal (V/H) ratios of seismic standards.

| Standards | V/H Ratio | |
|---|---|---|
| | High Frequency (Short Period) | Low Frequency (Long Period) |
| Eurocode 8 (Type 1) | 0.9 | 0.9 |
| Eurocode 8 (Type 2) | 0.45 | 0.45 |
| USNRC | 1 | 2/3 |
| ASCE 4–16 | 2/3 | 2/3 |
| NEHRP | 0.7 | 1/2 |

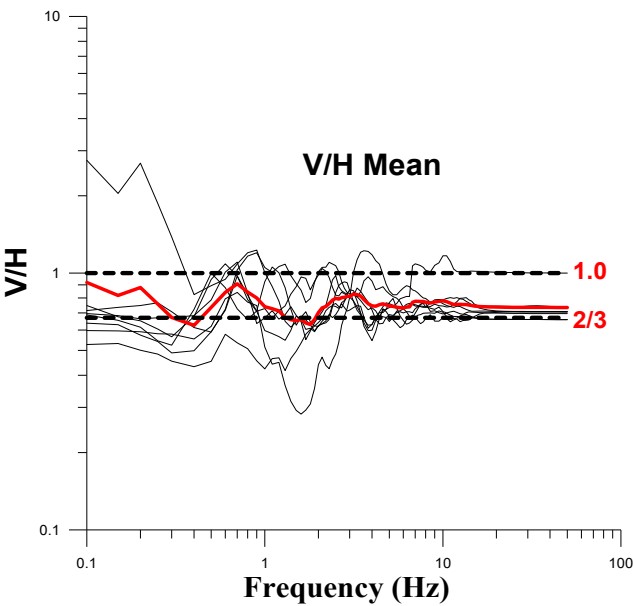

**Figure 9.** Mean and individual V/H ratios of the 8 seismic stations. The *x*-axis represents frequency and the *y*-axis the V/H ratios.

Moreover, this study also investigated the dependence of V/H values on the earthquake magnitude, as shown in Figure 10. The classification criteria of the three different levels of earthquake magnitude considered for this evaluation were the same as that in the previous section. Although some frequency bands were smaller than 3/2 in the V/H for all three seismic magnitude groups, they consistently showed values smaller than 1 in the entire frequency band and are independent of the magnitude of the earthquake, which is consistent with the various seismic design standards. Additionally, considering the vertical component is always smaller than the horizontal component response in the entire frequency band, no significant dependence of the V/H values on the earthquake magnitude for the horizontal and vertical components of the response spectra was observed.

### 5.5. Comparison of the Results with the Reg. Guide 1.60

As shown in Figure 11, we compared the mean and mean plus 1σ response spectra for the horizontal and vertical components of the eight seismic stations based on the ground motions observed in the Jeju region to that of the Reg. Guide 1.60, which mandates separate standards for vertical and horizontal ground motions and is used as the seismic design standard for nuclear power plants and related facilities in Korea, the US, and other nations.

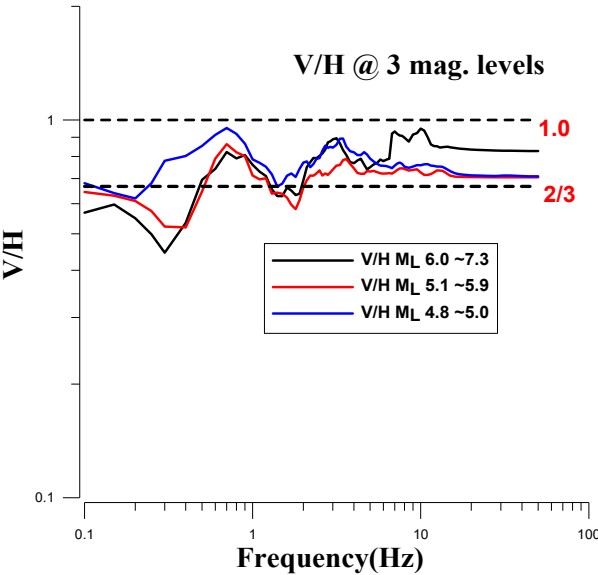

**Figure 10.** Mean V/H ratios of the 8 seismic stations for three groups of earthquake magnitude: $M_L$ 6.0–7.3 (black solid), 5.3–5.8 (red solid), and 4.6–5.0 (blue solid), respectively. The *x*-axis represents frequency and the *y*-axis the V/H ratios.

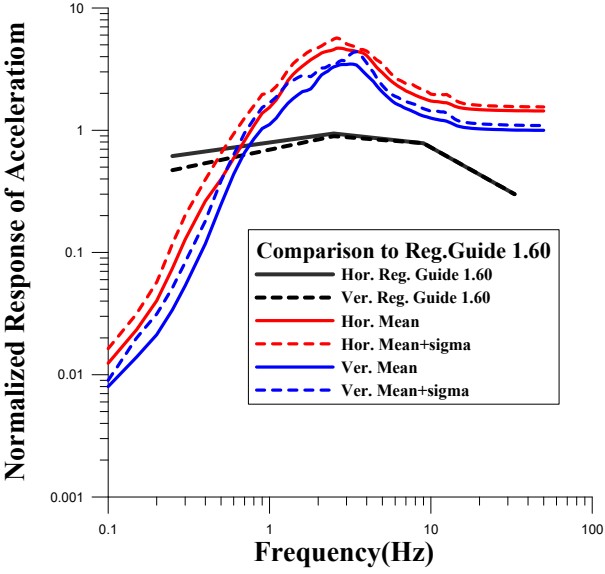

**Figure 11.** Comparison of normalised response spectra at 8 stations with the horizontal and vertical response spectra of Reg. 1.60 scaled to 0.3 g. The *x*-axis represents frequency and the *y*-axis is the amplification factor, equivalent to the response spectrum acceleration.

According to the internal regulations, the vertical spectra of Reg. Guide 1.60 are scaled to two-thirds that of the horizontal spectra. While we analyzed the response spectra of up to 50 Hz, Reg. Guide 1.60 only covered bands up to 33 Hz. To compare the observed response spectra to the design standards of Reg. Guide 1.60, both the horizontal and vertical spectra of Reg. Guide 1.60 were scaled down to 0.3 g, corresponding to the level of the design earthquake (the Safe Shutdown Earthquake [SSE]) of all Korean nuclear facilities.

The response spectra for both components were significantly lower than those of the design acceptance guidelines in the low-frequency band (0.6 Hz or less), indicating that the design standards are adequately conservative. However, for the frequency bands ranging from 0.6 Hz or higher, the response spectra for both components exceeded those of

Reg. Guide 1.60 scaled to 0.3g, indicating that the current seismic design standards for the specified frequency bands are not adequately conservative, particularly in the Jeju area.

In contrast, Kim and Oh (2016) from Fukuoka events series, Kim et al. (2018) from Pohang event series (ML5.4 or less), and Kim et al., (2019) from Gyoungju event series (ML 5.8 or less) demonstrate that the horizontal response spectra exceeded the seismic design standards with the same scale of 0.3 g and in frequency bands greater than ~7 Hz [14–16]. As suggested in the previous section, the exceeded response spectra in the frequency bands from 0.6 Hz or higher could be attributed to the seismo-tectonics and shallow crustal characteristics of the Jeju area. The higher responses than standard for broad ranges of frequencies are presumed to be caused by the large elastic impedance contrast between the fairly thick low-velocity marine sediment layer interposed in the shallow crust (Kim and Hong, 2012) and the basaltic/granitic bedrock (Choi, et at., 2007) in the Jeju island, however, more studies are needed [7,8].

Furthermore, compared to the design acceptance guidelines, the response spectra from all eight stations showed relatively stronger amplitudes in frequency bands greater than 25 Hz, which mainly affected small devices attached to walls having short resonance periods. Stable tectonic regions (Silva et al. 1999), such as the Korean Peninsula and EUS (Eastern United States), tend to have relatively stronger high-frequency responses compared to tectonically active regions such as WUS (Western United States) [22]. In particular, Reg. Guide 1.60 was developed based on tectonic recordings from active regions (primarily California). Therefore, it is natural that the normalized response spectra of the eight stations in this study showed stronger amplitudes than the nuclear standard in higher frequency bands. The response spectra exceeding the seismic design standards at high frequencies have also been reported in the nuclear power generation industry (Sun et al., 2007) [34].

For the construction of nuclear facilities in the Jeju area, the seismic design code for nuclear facilities Reg. Guide 1.60 should be carefully considered to provide appropriate safety margins for frequencies ranging from 0.6 Hz or higher including medium and high frequency bands. Therefore, instead of relying on Reg. Guide 1.60, it is necessary to significantly improve the current design standards or develop new site-specific seismic design standards for nuclear facilities that reflect the local seismo-tectonic and geological environment of the Jeju region.

*5.6. Comparison with KBC 2016*

We compared the mean and mean plus 1σ response spectra of the eight seismic stations for the horizontal components (vector sum of NS and EW components) in the Jeju region to those of KBC 2016, which is used as the seismic design standard in Korea for general buildings and structures such as hospitals and schools. In contrast with Reg. Guide 1.60, KBC 2016 does not distinguish between the vertical and horizontal spectra, and the design response spectrum of KBC 2016 considers both the short- ($S_{DS}$) and long-period ($S_{D1}$) seismic hazards based on site amplification of Fa (short) and Fv (long), respectively.

As shown in Figure 12, we considered three categories of KBC 2016 design response spectra based on the soil type (SC, SD, and SE), following the processes suggested in the KBC 2016 codes. Site soil types—SC, SD, and SE—were selected considering they cover most soils in Korea. Figure 12 shows the response spectra corresponding to the soil types SC, SD, and SE in the order of distance from the x-axis. Based on an example in KBC 2016, a return period of 2400 years, which is determined by considering the importance of buildings and structures and the degree of socially mandated performance levels, was selected for this study. As the level of amplification of the ground increases in the order of SC, SD, and SE, the acceptance guideline of design hazard levels increases in the same order.

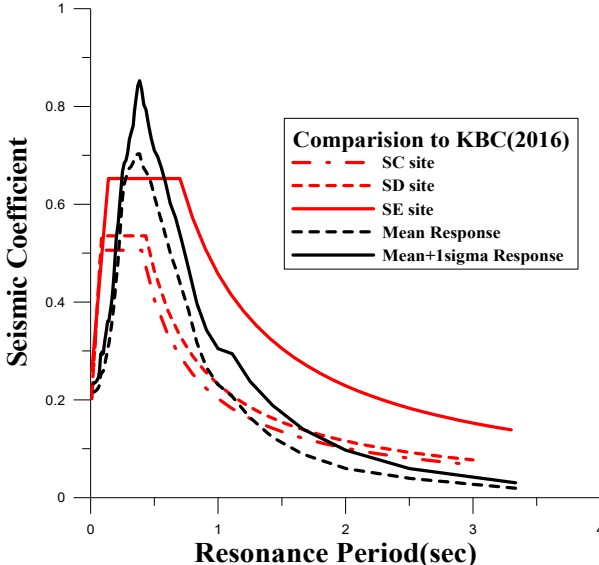

**Figure 12.** Comparison of the response spectra of 8 stations with KBC (2016) for SC, SD, and SE soil conditions (2400-years return periods). The *x*- and *y*-axes represent the structure resonance period and response spectrum, respectively.

For soil SE, the horizontal response spectra of the eight stations for mean plus 1σ were exceeded by a fairly large margin (~0.25–0.60 s or ~1.33–4 Hz) compared to the design standard. On comparing the mean response spectra, it was observed that the range exceeding soil SE becomes narrow naturally, as shown in Figure 12.

We obtained in this section the design response spectra by comprehensively considering both seismic coefficient and site amplification (Fa and Fv), following the regulation process (0306.3.3) by KBC 2016. In contrast, a simple calculation using the formula for a brief method (0306.5.6) suggested also by KBC 2016 was conducted in the previous section.

The period of exceeding the design response spectra (~0.25–0.60 s) in this section corresponded to the resonance period of 2–7 story buildings from the brief formula (0306.5.6) by KBC 2016. This implies that the re-establishment of the general seismic design standards of buildings, including the site-specific design response spectra, is required in the Jeju region.

## 6. Conclusions

In this study, we investigated the response spectra of eight stations for horizontal and vertical components in the Jeju area and explored the effects of earthquake magnitude on the resonance frequencies of buildings and other structures. Furthermore, we compared the response spectra of the eight stations to that of Reg. Guide 1.60 (anchored to 0.3 g) and KBC 2016 (2400-year return period) seismic design standards. The results can be summarized as follows:

1.  We analyzed 24 macro-earthquakes, including the Kumamoto main shock (2016-04-16, ML; 7.3) and Pohang main shock (2017-11-15, ML; 5.4). Their response spectra showed characteristics similar to the typical design response spectrum shapes, as suggested by the various design standards, Reg Guide 1.60, 1.165, and 1.208 (US NRC 1973, 1997, 2007) and ASME (Gupta and Hall, 2017) [10,17,20,21].
2.  The vertical response spectra of the eight stations were consistently lower than the vector-summed horizontal spectra, which was in good agreement with the V/H scales suggested by various seismic design standards (Table 3). Furthermore, the vertical response spectra clearly indicated magnitude dependence in the lower frequency band.
3.  When establishing design response spectra that represents an arbitrary region or the Jeju region, it is necessary to consider a number of fairly large magnitudes of earthquakes (ML 6.0 or larger) considering the earthquake magnitude is essential for determining the characteristics of the response spectrum, especially in the low-frequency

bands suggested in this and other studies. Fairly large earthquake magnitudes, e.g., $M_L$ 6.0 or larger, secure conservatism in the low-frequency ranges suggested by this study and other studies and are also important to reduce variability of observed response spectra. Considering the rare occurrence of seismic activities of ML 6.0 or larger in the Korean Peninsula, it is necessary to collect seismic ground motions from the south-eastern region of Japan, which is geographically closer to the Korean Peninsula, where earthquakes of ML 6.0 or larger occur frequently.

4.  Compared to that of inland areas, the maximum horizontal response in the Jeju area was fairly higher (~12%–13 %), whereas the frequency band with high responses at the peak (~2.0–6.0 Hz) was significantly lower. This could be attributed to the local geological environment of Jeju Island, which was formed mainly by a surface volcano eruption with lower shear velocity and is related to the existence of unconsolidated Pleistocene marine sediments distributed widely in the area. Further studies are necessary to understand the characteristic influence of layers with low shear wave velocity distributed in the Jeju region on the seismic responses based on the frequency band and amplitudes at the surface of Jeju.

5.  In this study, most frequency ranges with very high responses, including the peak, overlapped with the resonance frequencies of general low-rise and mid-rise buildings (2-story~7 story) considering both the brief formula and design response spectra suggested by KBC 2016, in contrast to frequencies higher than ~ 9 Hz in inland areas, which is far from the resonance frequency band. Owing to this, the seismic hazard on general buildings in the Jeju region could be much higher than that in inland areas. Therefore, it is necessary to establish an improved design response spectrum with site-specific response spectra in Jeju or to alter existing standards by applying natural resonance frequency ranges that are different from that of the inland areas.

6.  For the construction of nuclear facilities in the Jeju area, the seismic design code for nuclear facilities as suggested in Reg. Guide 1.60, should be carefully considered to provide the appropriate safety margins for frequencies ranging 0.6 Hz or higher. Therefore, rather than relying only on Reg. Guide 1.60, it is recommended to significantly improve the current design standards or develop new site-specific seismic design standards for nuclear facilities, which reflect the local seismo-tectonic and geological environment.

**Author Contributions:** J.-K.K., S.-H.W. and S.-H.Y. equally contributed to the research conceptualization and all investigations performed under its purview. Resource procurement was handled by J.-K.K. and S.-H.W. The original draft of the manuscript was prepared by J.-K.K. and S.-H.W., whereas the final draft was reviewed and edited by J.-K.K., S.-H.W. and K.-H.K. All authors have read and agreed to the published version of the manuscript.

**Funding:** This study was supported by the Korea Meteorological Administration Research and Development Program under Grant KMI2018-02810.

**Institutional Review Board Statement:** Not applicable.

**Informed Consent Statement:** Not applicable.

**Conflicts of Interest:** The authors declare no conflict of interest. The funders had no role in the design of the study; in the collection, analyses, or interpretation of data; in the writing of the manuscript, or in the decision to publish the results.

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
