# Peer review of "Characteristics of the Vertical and Horizontal Response Spectra of Earthquakes in the Jeju Island Region"

_applsci, doi:10.3390/app112210690_

Round 1

Reviewer 1 Report

The manuscript explores the resonance frequencies of structures and buildings due to the effects of earthquakes magnitudes.

Also, it evaluates response spectra of 24 earthquakes and compares them with the design spectra Regulatory Guide 1.60.

The introduction is well structured and the references are relevant.

Figure 1 must add the names of the regions and also the tectonics.

The five first subsections of section 4 should go to another new section with methodology or processing data.

The Conclusions are concreted and supported by the results.

In general an exciting manuscript.

Also, I include a file with some typos.

Reviewer 2 Report

Presented manuscript evaluated the response spectra of 24 EQ series in the Jeju Island Region and explored the effects of EQ magnitude on the resonance frequencies of structures and buildings and observations were compared with the design response spectra.

Main complaint about manuscript is a lot of self-referencing (Kim). MDPI reference style is not used. Manuscript is written poorly with a lot of copy-paste sentences from other sources without proper referencing. English needs to be checked. 

My review is as stated:

  • Lines 40-43 - please rewrite, what many studies?
  • Lines 44-45 - can you show seismicity map?
  • Line 58 - Figure 1 is mentioned yet, Figure 1 is on page 4/19 and shows Kumamoto and Pohang EQ series.
  • Lines 56-65 - MDPI reference style. 
  • Lines 66-73 - proper referencing needed here.
  • Present Figure 1 as presented in Table 1, circles with magnitude range and legend. This is unproper seismicity map. Please adjust Figure 1 - in my opinion is too big in coordinate range. Where is Jeju Island on the map?
  • In chapter 3, where did you takend Eq.(1)? Any reference to that? 
  • Again, lines 161-164 are taken from somewhere? You cannot state these things, while there are so many studies and books on that topic? 
  • In Chapter 4, how did you produce Response spectra from ground motions? Which software did you used? What lines state for? Which earthquake?
  • Lines 218-221 - how did you produce this number? According to which equation? This kind of facts needs to be explained or at least point to reference.
  • How is possible that Figure 2 is shown in line 272 while prior there were 4 figures? I must state that manuscript is hard to read, a lot of misconfusing parts...
  • I think there exists a lot of references on the magnitude dependency on the response spectra, yet none of this were mentioned, e.g. prof. Boore articles on the Fourier spectra, Response spectra, etc. I would recommend to study novel literature on thi topic, more importantly on the Response Spectra, the re-write whole study. Three groups in Figure 2 (probably 5) cannont be put into the same plot? Are epicentral distances the same? What about local site effects? What about source effects? Whole this part is crucial point, yet confusingly written with no-presented crucial points.
  • Lines 392-404 - what about other references on this topic? Other studies? I would not agree that 5-story builiding (RC) has 2 Hz? This is simply not true! 10-15 story building has about 2 Hz!
  • LInes 407-410- while whole world uses H/V ratio, you used V/H ratio and stated "many seismic design criteria". Why not show apperantly also H/V ratio to see distinction compare to V/H ratio? Then this can be compared to buildings resonance.
  • In Figure 9, I am not sure that black line is correct - please check? If you scaled this ot 0.3g, what scale did you use for red and blue spectra? They should also be scaled to this pga then.
  • In Fig 10, why not compare to other design spectra mentioned in Table 3 if you adopted V/H ratio from them.
  • LIne 512 - I do not understand, 2400 year return period? Why?! This should be explained, otherwise show it for 475.
  • Line 514 - how level of amplification is determined?
  • Line 547 - what do you mean by macro EQ? Macro in seismology means something else?

I am sorry, but I recommend that manuscript is thourghly checked and re-written. Suggestion - when you present some facts they need to be supported by reference or somehow, not just write as your own fact.  Also a lot of confusing terminology.

Reviewer 3 Report

In this paper, the characteristics of seismic motion on Jeju Island are considered based on the evaluation of the normalized response spectra of the seismic motion data observed on Jeju Island. Jeju Island is a region with low seismic activity, and it is difficult to evaluate the strong ground motion caused by a nearby earthquake. For this reason, the authors are analyzing two earthquakes that occurred at a distance.

Although the analysis is limited to the data that can be used, it is recognized as a basic material for earthquake disaster prevention measures on Jeju Island. The issues are clearly organized, and it can be judged that the paper is worth publishing.

Since only the normalized response spectra are shown in the paper, the data used in the discussion is lacking in information. It is desirable to make corrections to the following points.

  1. The time history waveforms of the records of the mainshocks and typical aftershocks for the Kumamoto earthquake series and the Pohang earthquake series for the representative points should be shown in figures.

2. The response spectra of the mainshock and typical aftershocks for two earthquake series for representative points are shown in figures.

Author Response

The corrections were reflected in the revised manuscript.

Round 2

Reviewer 1 Report

The manuscript was corrected.

Good job authors.

Author Response

Thank you for your comments and suggestions.

Reviewer 2 Report

In the  second review authors still did not correct reference style?! You cannot use both styles!

Figure 2 - font is too small.

Author Response

The manuscript has been revised according to comments and suggestions.
Thank you for your review.